# Meta-Learning for Relative Density-Ratio Estimation

**Atsutoshi Kumagai**
NTT Computer and Data Science Laboratories
atsutoshi.kumagai.ht@hco.ntt.co.jp

**Tomoharu Iwata**
NTT Communication Science Laboratories
tomoharu.iwata.gy@hco.ntt.co.jp

**Yasuhiro Fujiwara**
NTT Communication Science Laboratories
yasuhiro.fujiwara.kh@hco.ntt.co.jp

## Abstract

The ratio of two probability densities, called a density-ratio, is a vital quantity in machine learning. In particular, a relative density-ratio, which is a bounded extension of the density-ratio, has received much attention due to its stability and has been used in various applications such as outlier detection and dataset comparison. Existing methods for (relative) density-ratio estimation (DRE) require many instances from both densities. However, sufficient instances are often unavailable in practice. In this paper, we propose a meta-learning method for relative DRE, which estimates the relative density-ratio from a few instances by using knowledge in related datasets. Specifically, given two datasets that consist of a few instances, our model extracts the datasets' information by using neural networks and uses it to obtain instance embeddings appropriate for the relative DRE. We model the relative density-ratio by a linear model on the embedded space, whose global optimum solution can be obtained as a closed-form solution. The closed-form solution enables fast and effective adaptation to a few instances, and its differentiability enables us to train our model such that the expected test error for relative DRE can be explicitly minimized after adapting to a few instances. We empirically demonstrate the effectiveness of the proposed method by using three problems: relative DRE, dataset comparison, and outlier detection.

## 1 Introduction

The ratio of two probability densities, called a density-ratio, has been used in various applications such as outlier detection [14, 1], dataset comparison [49], covariate shift adaptation [42], change point detection [30], positive and unlabeled (PU) learning [19, 18], density estimation [46], and generative adversarial networks [47]. Thus, density-ratio estimation (DRE) is attracting a lot of attention. A naive approach to DRE is to estimate each density and then take the ratio. However, this approach does not work well since density estimation is a hard problem [48]. Therefore, direct DRE without going through density estimation has been extensively studied [44, 17, 35, 13].

Although direct DRE is useful, its fundamental weakness is that the density-ratio is unbounded, i.e., it can take infinity, which causes stability issues [29]. To cope with this problem, a relative density-ratio has been proposed, which is a smoothed and bounded extension of the density-ratio [49]. In the above applications, the density-ratio can be replaced with the relative density-ratio, and relative DRE has shown excellent performance [49, 30, 6, 37, 47].

Existing methods for (relative) DRE require many instances from both densities. However, sufficient instances are often unavailable for various reasons. For example, it is difficult to instantly collect many instances from new data sources such as new users or new systems. Collecting instances is

expensive in some applications such as clinical trials or crash tests, where DRE can be used for dataset comparison to investigate the effect of drugs/car conditions. In such cases, existing methods cannot work well.

In this paper, we propose a meta-learning method for relative DRE. To estimate the relative density-ratio from a few instances in target datasets, the proposed method uses instances in different but related datasets, called source datasets. When these datasets are related, we can transfer useful knowledge from source datasets to target ones [31]. Figure 1 shows our problem formulation.

We model the relative density-ratio by using neural networks that enable us to perform accurate DRE thanks to their high expressive capabilities. Since each dataset has a different property, incorporating it to the model is essential. To achieve this, given two datasets that consist of a few instances, called support instances, our model first calculates a latent vector representation of each dataset. This vector is calculated by using permutation-invariant neural networks that can take a set of instances as input [50]. Since the vector is obtained from a set of instances in the dataset, it contains information of the dataset.

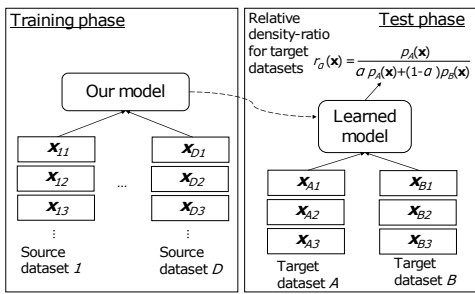

Figure 1: Our problem formulation. In a training phase, our model is trained with source datasets. In a test phase, the learned model estimates relative density-ratio $r_\alpha(\mathbf{x}) = \frac{p_A(\mathbf{x})}{\alpha p_A(\mathbf{x}) + (1-\alpha) p_B(\mathbf{x})}$, $(0 \leq \alpha < 1)$ with target datasets $A$ and $B$ that are generated from densities $p_A(\mathbf{x})$ and $p_B(\mathbf{x})$, respectively.

With the two latent vectors of datasets, each instance is non-linearly mapped to an embedding space that is suitable for relative DRE on the datasets. Using the embedded instances, we perform relative DRE, where the relative density-ratio is represented by a linear model on the embedded space. With the squared loss, the global optimal solution of the linear model can be obtained as a closed-form solution, which enables us to perform more stable and faster adaptation to support instances than numerical solutions.

The neural networks of our model are trained by minimizing the expected test squared error of relative DRE after adapting to support instances that is calculated using instances in the source datasets. Since the closed-form solution of the linear model is differentiable, this training can be performed by gradient-based methods such as ADAM [20]. Since all parameters of our model are shared across all datasets, which enables knowledge to be shared between all datasets, the learned model can be applied to unseen target datasets. This training explicitly improves the relative DRE performance for test instances after estimating the relative density-ratio using support instances. Thus, the learned model can accurately estimate the relative density-ratio from a few instances.

Our main contributions are as follows: (1) To the best of knowledge, our work is the first attempt at meta-learning for (relative) DRE. (2) We propose a model that performs accurate relative DRE from a few instances by effectively adapting both embeddings and linear model to the instances. (3) We empirically demonstrate the effectiveness of the proposed method with three popular problems: relative DRE, dataset comparison, and outlier detection.

## 2   Related Work

Many methods for direct DRE have been proposed such as classifier-based methods [4, 33], Kullback–Leibler importance estimation [43], kernel mean matching [13], and unconstrained least-squares importance fitting (uLSIF) [17]. Although these methods are useful, they can suffer from the unbounded nature of the density-ratio. That is, an instance that is in the low density region of the denominator density may have an extremely large value of ratio, and DRE can be dominated by such points, which causes robustness and stability issues [29]. This problem is particular serious when using flexible density-ratio models such as neural networks because they try to fit on extremely large density-ratio values [18]. To cope with the unboundedness, the relative uLSIF (RuLSIF) uses the relative density-ratio to uLSIF [49]. Since RuLSIF can obtain the optimal parameters as a closed-form solution, it is computationally efficient and stable and performs well in various applications when sufficient instances are available [49, 30, 6, 37, 47]. Since fast and effective adaptation to support in-

stances is essential in meta-learning as explained in the end of this section, we incorporate RuLSIF in our framework. Although neural network-based DRE methods have been recently proposed [18, 35], they cannot perform well when training instances are quite small due to overfitting. Although the proposed method uses neural networks for the instance embeddings, it can accurately perform relative DRE from a few instances by learning how to perform few-shot relative DRE with related datasets.

DRE is used for transfer learning or covariate shift adaptation [40, 42, 45, 22, 7]. To transfer knowledge in a training dataset to a test dataset, these methods estimate the density-ratio between training and test densities that is used for weighting labeled training instances. These methods use only two datasets to estimate the density-ratio. In contrast, the proposed method uses multiple datasets to accumulate transferable knowledge and uses it for the relative DRE on two new datasets, which can be used in various applications including covariate shift adaptation as described in Section 1.

Meta-learning methods have been recently attracting a lot of attention [8, 41, 9, 34, 3, 16]. These methods train a model such that it generalizes well after adapting to support instances using multiple datasets. In this framework, fast adaptation to support instances is essential since the result of the adaptation is required to train the model in each iteration of training [3, 34]. Encoder-decoder methods such as neural processes [9, 10] perform quick adaptation by forwarding support instances to neural networks. However, they have difficulty working well for any dataset since the adaptation is approximated by only the neural networks. Gradient-based methods such as model-agnostic meta-learning (MAML) [8] adapt to support instances by using an iterative gradient descent method and are widely used. These methods require higher-order derivatives and to retain all optimization path of the iterative adaptation to backpropagate through the path, which imposes considerable computational and memory burdens [3]. Thus, they must keep the iteration number small and it prevents effective adaptation. In contrast, the proposed method quickly and effectively adapt to support instances by solving a convex optimization problem, where the global optimum solution can be quickly obtained as a closed-form solution. Although few methods adapt to support instances by solving convex optimization problems for fast and effective adaptation [3, 26], they consider classification tasks. To the best of our knowledge, no meta-learning methods have been designed for DRE, and thus, existing meta-learning methods cannot be applied to our problems.

# 3   Preliminary

We briefly explain a relative density-ratio. Suppose that instances $\{\mathbf{x}_n\}_{n=1}^N$ are drawn from a distribution with density $p_{\mathrm{nu}}(\mathbf{x})$ and instances $\{\mathbf{x}_n'\}_{n=1}^{N'}$ are drawn from another distribution with density $p_{\mathrm{de}}(\mathbf{x})$. Density ratio $r(\mathbf{x})$ is defined by $r(\mathbf{x}) := \frac{p_{\mathrm{nu}}(\mathbf{x})}{p_{\mathrm{de}}(\mathbf{x})}$. Here, "nu" and "de" indicate the numerator and the denominator. The aim of DRE is to directly estimate $r(\mathbf{x})$ from both instances $\{\mathbf{x}_n\}_{n=1}^N$ and $\{\mathbf{x}_n'\}_{n=1}^{N'}$. However, $r(\mathbf{x})$ is unbounded and thus can take extremely large values when the denominator $p_{\mathrm{de}}(\mathbf{x})$ takes a small value. This causes robustness and stability issues [29]. To deal with this problem, a relative density-ratio has been proposed [49]. For $0 \le \alpha < 1$, relative density-ratio $r_\alpha(\mathbf{x})$ is define by $r_\alpha(\mathbf{x}) := \frac{p_{\mathrm{nu}}(\mathbf{x})}{\alpha p_{\mathrm{nu}}(\mathbf{x}) + (1-\alpha)p_{\mathrm{de}}(\mathbf{x})}$. Relative density-ratio $r_\alpha(\mathbf{x})$ is bounded since $r_\alpha(\mathbf{x}) \le \frac{1}{\alpha}$ for any $\mathbf{x}$. $r_\alpha(\mathbf{x})$ is always smoother than $r(\mathbf{x})$. $r_\alpha(\mathbf{x})$ can replace $r(\mathbf{x})$ and is used for various applications [49, 30, 6, 37, 47]. When $\alpha = 0$, $r_\alpha(\mathbf{x})$ is reduced to density-ratio $r(\mathbf{x})$. Thus, the relative density-ratio can be regarded as a smoothed and bounded extension of the density-ratio.

# 4   Proposed Method

## 4.1   Problem Formulation

Let $X_d = \{\mathbf{x}_{dn}\}_{n=1}^{N_d}$ be a $d$-th dataset and $\mathbf{x}_{dn} \in \mathbb{R}^M$ be the $M$-dimensional feature vector of $n$-th instance in the $d$-th dataset. Instances $\{\mathbf{x}_{dn}\}_{n=1}^{N_d}$ are drawn from a distribution with density $p_d$. We assume that feature dimension $M$ is the same across all datasets, but each distribution can differ. Suppose that $D$ datasets $X := \{X_d\}_{d=1}^D$ are given at the training phase. Our goal is to estimate a relative density-ratio $r_\alpha(\mathbf{x})$ from two target datasets that consist of a few instances, $\mathcal{S}_{d_{\mathrm{nu}}} = \{\mathbf{x}_{d_{\mathrm{nu}}n}\}_{n=1}^{N_{d_{\mathrm{nu}}}}$ and $\mathcal{S}_{d_{\mathrm{de}}} = \{\mathbf{x}_{d_{\mathrm{de}}n}\}_{n=1}^{N_{d_{\mathrm{de}}}}$, where $d_{\mathrm{nu}}$ and $d_{\mathrm{de}}$ are not included in $\{1 \ldots, D\}$, that are given at the test phase.

## 4.2 Model

In this subsection, we use notations $\mathcal{S}_{\mathrm{nu}}$ and $\mathcal{S}_{\mathrm{de}}$ instead of $\mathcal{S}_{d_{\mathrm{nu}}}$ and $\mathcal{S}_{d_{\mathrm{de}}}$, respectively for simplicity. Similarly, we use $p_{\mathrm{nu}}$ and $p_{\mathrm{de}}$ instead of $p_{d_{\mathrm{nu}}}$ and $p_{d_{\mathrm{de}}}$, respectively. We explain our model that estimates the relative density-ratio from $\mathcal{S} = \mathcal{S}_{\mathrm{nu}} \cup \mathcal{S}_{\mathrm{de}}$, called support instances. First, our model calculates a latent representation of each dataset using permutation-invariant neural networks [50]:

$$\mathbf{z}_{\mathrm{nu}} := g\left(\frac{1}{|\mathcal{S}_{\mathrm{nu}}|}\sum_{\mathbf{x}\in\mathcal{S}_{\mathrm{nu}}} f(\mathbf{x})\right) \in \mathbb{R}^K, \quad \mathbf{z}_{\mathrm{de}} := g\left(\frac{1}{|\mathcal{S}_{\mathrm{de}}|}\sum_{\mathbf{x}\in\mathcal{S}_{\mathrm{de}}} f(\mathbf{x})\right) \in \mathbb{R}^K, \qquad (1)$$

where $f$ and $g$ are any feed-forward neural network. Since summation is permutation-invariant, the neural network in Eq. (1) outputs the same vector even though the order of instances in each dataset varies. Thus, the neural network in Eq. (1) is well defined as functions for set inputs. Since latent vector $\mathbf{z}$ is calculated from the set of instances in a dataset, $\mathbf{z}$ contains information of the empirical distribution of instances in the dataset. The proposed method can use any other permutation-invariant function such as summation [50] and set transformer [25] to obtain latent vectors of datasets.

The proposed method models the relative density-ratio by the following neural network,

$$\hat{r}_\alpha(\mathbf{x}; \mathcal{S}) := \mathbf{w}^\top h([\mathbf{x}, \mathbf{z}_{\mathrm{nu}}, \mathbf{z}_{\mathrm{de}}]), \qquad (2)$$

where $[\cdot, \cdot, \cdot]$ is a concatenation of vectors, $h : \mathbb{R}^{M+2K} \to \mathbb{R}^T_{\geq 0}$ is a feed-forward neural network, and $\mathbf{w} \in \mathbb{R}^T_{\geq 0}$ is linear weights. The non-negativeness of both the outputs of $h$ and $\mathbf{w}$ ensures the non-negativeness of the estimated relative density-ratio. $h([\mathbf{x}, \mathbf{z}_{\mathrm{nu}}, \mathbf{z}_{\mathrm{de}}])$ represents the embedding of instance $\mathbf{x}$. Since $h([\mathbf{x}, \mathbf{z}_{\mathrm{nu}}, \mathbf{z}_{\mathrm{de}}])$ depends on both $\mathbf{z}_{\mathrm{nu}}$ and $\mathbf{z}_{\mathrm{de}}$, the embeddings reflect the characteristics of two datasets. Such embeddings are learned by using source datasets $X$ so that they lead to accurate DRE given the target datasets, which will be described in the subsection 4.3.

Linear weights $\mathbf{w}$ are determined so that the following expected squared error between true relative density-ratio $r_\alpha(\mathbf{x})$ and estimated relative density-ratio $\hat{r}_\alpha(\mathbf{x}; \mathcal{S})$, $J_\alpha$, is minimized:

$$J_\alpha(\mathbf{w}) := \frac{1}{2}\mathbb{E}_{q_\alpha(\mathbf{x})}\left[(r_\alpha(\mathbf{x}) - \hat{r}_\alpha(\mathbf{x}; \mathcal{S}))^2\right]$$
$$= \frac{\alpha}{2}\mathbb{E}_{p_{\mathrm{nu}}(x)}\left[\hat{r}_\alpha(\mathbf{x}; \mathcal{S})^2\right] + \frac{1-\alpha}{2}\mathbb{E}_{p_{\mathrm{de}}(x)}\left[\hat{r}_\alpha(\mathbf{x}; \mathcal{S})^2\right] - \mathbb{E}_{p_{\mathrm{nu}}(x)}\left[\hat{r}_\alpha(\mathbf{x}; \mathcal{S})\right] + \mathrm{Const.}, \quad (3)$$

where $\mathbb{E}$ is expectation, $q_\alpha(\mathbf{x}) := \alpha p_{\mathrm{nu}}(\mathbf{x}) + (1-\alpha)p_{\mathrm{de}}(\mathbf{x})$, and $\mathrm{Const}$ is a constant term that does not depend on our model. By approximating the expectation with support instances $\mathcal{S}$ and excluding the non-negative constraints for $\mathbf{w}$, we obtained the following optimization problem:

$$\tilde{\mathbf{w}} := \underset{\mathbf{w}\in\mathbb{R}^T}{\arg\min}\left[\frac{1}{2}\mathbf{w}^\top \mathbf{K}\mathbf{w} - \mathbf{k}^\top \mathbf{w} + \frac{\lambda}{2}\mathbf{w}^\top \mathbf{w}\right], \qquad (4)$$

where $\mathbf{k} = \frac{1}{|\mathcal{S}_{\mathrm{nu}}|}\sum_{\mathbf{x}\in\mathcal{S}_{\mathrm{nu}}} h([\mathbf{x}, \mathbf{z}_{\mathrm{nu}}, \mathbf{z}_{\mathrm{de}}])$ and $\mathbf{K} = \frac{\alpha}{|\mathcal{S}_{\mathrm{nu}}|}\sum_{\mathbf{x}\in\mathcal{S}_{\mathrm{nu}}} h([\mathbf{x}, \mathbf{z}_{\mathrm{nu}}, \mathbf{z}_{\mathrm{de}}])h([\mathbf{x}, \mathbf{z}_{\mathrm{nu}}, \mathbf{z}_{\mathrm{de}}])^\top + \frac{1-\alpha}{|\mathcal{S}_{\mathrm{de}}|}\sum_{\mathbf{x}\in\mathcal{S}_{\mathrm{de}}} h([\mathbf{x}, \mathbf{z}_{\mathrm{nu}}, \mathbf{z}_{\mathrm{de}}])h([\mathbf{x}, \mathbf{z}_{\mathrm{nu}}, \mathbf{z}_{\mathrm{de}}])^\top$. In Eq. (4), the third term of r.h.s. is the $\ell^2$-regularizer to prevent over-fitting, and $\lambda > 0$ is a positive real number. The global optimum solution for Eq. (4) can be obtained as the following closed-form solution:

$$\tilde{\mathbf{w}} = (\mathbf{K} + \lambda\mathbf{I})^{-1}\mathbf{k}, \qquad (5)$$

where $\mathbf{I}$ is the $T$ dimensional identity matrix. This closed-form solution can be efficiently obtained when $T$ is not large. Note that $(\mathbf{K} + \lambda\mathbf{I})^{-1}$ exists since $\lambda > 0$ makes $(\mathbf{K} + \lambda\mathbf{I})$ positive-definite. Some learned weights $\tilde{\mathbf{w}}$ can be negative. To compensate for this, following previous studies [17], the solution is modified as $\hat{\mathbf{w}} = \max(0, \tilde{\mathbf{w}})$, where $\max$ operator is applied for each element of $\tilde{\mathbf{w}}$. The closed-form solution enables fast and effective adaptation to support instances $\mathcal{S}$. By using the learned weights, the relative density-ratio estimated with support instances $\mathcal{S}$ can be obtained as

$$\hat{r}_\alpha^*(\mathbf{x}; \mathcal{S}) := \hat{\mathbf{w}}^\top h([\mathbf{x}, \mathbf{z}_{\mathrm{nu}}, \mathbf{z}_{\mathrm{de}}]). \qquad (6)$$

## 4.3 Training

We explain the training procedure for our model. In this subsection, symbols $\mathcal{S} = \mathcal{S}_{\mathrm{nu}} \cup \mathcal{S}_{\mathrm{de}}$ are used as support instances in source datasets. In our model, the parameters to be estimated, $\Theta$, are

---

**Algorithm 1** Training procedure of our model.

---

**Require:** Source datasets $X$, support instance size $N_{\mathcal{S}}$, query instance size $N_{\mathcal{Q}}$, relative parameter $\alpha$
**Ensure:** Parameters of our model $\Theta$
1: **repeat**
2:     Sample two datasets $d$ and $d'$ from $\{1, \ldots, D\}$ with replacement
3:     Select support instances $\mathcal{S}_{\mathrm{nu}}$ and $\mathcal{S}_{\mathrm{de}}$ with size $N_{\mathcal{S}}$ from $X_d$ and $X_{d'}$, respectively
4:     Select query instances $\mathcal{Q}_{\mathrm{nu}}$ and $\mathcal{Q}_{\mathrm{de}}$ with size $N_{\mathcal{Q}}$ from $X_d$ and $X_{d'}$, respectively
5:     Calculate linear weights $\tilde{\mathbf{w}}$ with the support instances by Eq. (5) to obtain Eq. (6)
6:     Calculate the loss $\tilde{J}_\alpha(\mathcal{Q}; \mathcal{S})$ in Eq. (8) with the query instances
7:     Update parameters with the gradients of the loss $\tilde{J}_\alpha(\mathcal{Q}; \mathcal{S})$
8: **until** End condition is satisfied;

---

neural network parameters $f$, $g$, $h$, and regularizer parameter $\lambda$. We estimate these parameters by minimizing the expected test squared error of relative DRE given support instances, where support instances $\mathcal{S} = \mathcal{S}_{\mathrm{nu}} \cup \mathcal{S}_{\mathrm{de}}$ and test instances $\mathcal{Q} = \mathcal{Q}_{\mathrm{nu}} \cup \mathcal{Q}_{\mathrm{de}}$, called query instances, are randomly generated from source datasets $X$:

$$\mathbb{E}_{d,d'\sim\{1,\ldots,D\}}\left[\mathbb{E}_{(\mathcal{S}_{\mathrm{nu}},\mathcal{S}_{\mathrm{de}}),(\mathcal{Q}_{\mathrm{nu}},\mathcal{Q}_{\mathrm{de}})\sim X_d\times X_{d'}}\left[\tilde{J}_\alpha(\mathcal{Q};\mathcal{S})\right]\right], \qquad (7)$$

where $(U, V) \sim X_d \times X_{d'}$ denotes that instances $U$ and $V$ are selected from $X_d$ and $X_{d'}$, respectively, and $\tilde{J}_\alpha(\mathcal{Q}; \mathcal{S})$ is the approximation of expected squared error $J_\alpha$ with query instances $\mathcal{Q}$,

$$\tilde{J}_\alpha(\mathcal{Q};\mathcal{S}) = \frac{\alpha}{2|\mathcal{Q}_{\mathrm{nu}}|}\sum_{\mathbf{x}\in\mathcal{Q}_{\mathrm{nu}}}\hat{r}_\alpha^*(\mathbf{x};\mathcal{S})^2 + \frac{1-\alpha}{2|\mathcal{Q}_{\mathrm{de}}|}\sum_{\mathbf{x}\in\mathcal{Q}_{\mathrm{de}}}\hat{r}_\alpha^*(\mathbf{x};\mathcal{S})^2 - \frac{1}{|\mathcal{Q}_{\mathrm{nu}}|}\sum_{\mathbf{x}\in\mathcal{Q}_{\mathrm{nu}}}\hat{r}_\alpha^*(\mathbf{x};\mathcal{S}). \quad (8)$$

The pseudocode for our training procedure is illustrated in Algorithm 1. For each iteration, we randomly select two datasets with replacement (Line 2). From the datasets, we randomly select support instances $\mathcal{S} = \mathcal{S}_{\mathrm{nu}} \cup \mathcal{S}_{\mathrm{de}}$ and query instances $\mathcal{Q} = \mathcal{Q}_{\mathrm{nu}} \cup \mathcal{Q}_{\mathrm{de}}$ (Lines 3–4). We then calculate the relative density-ratio with the support instances (Line 5). Using the estimated relative density-ratio, we calculate loss $\tilde{J}_\alpha(\mathcal{Q}; \mathcal{S})$ with the query instances (Line 6). Lastly, the parameters of our model are updated with the gradient of the loss (Line 7). This training procedure trains the parameters of our model so as to explicitly improve the relative DRE performance after estimating the relative density-ratio with a few instances. Thus, the learned model makes accurate DRE from target support instances. Since the close-form solution for adaptation in Eq. (5) is differentiable w.r.t. the model parameters, this training can be performed by using gradient-based methods such as ADAM [20] . Although we use the squared error for the objective function of query instances in Eq. (8), our framework can use any differentiable loss function for query instances, such as KullbackLeibler divergence [43] since we do not require closed-form solutions for query instances. A more intuitive explanation of the proposed method is described in the supplemental material.

## 5 Experiments

In this section, we demonstrate the effectiveness of the proposed method with three problems: relative DRE, dataset comparison, and inlier-based outlier detection. All experiments were conducted on a Linux server with an Intel Xeon CPU and a NVIDIA GeForce GTX 1080 GPU.

### 5.1 Proposed Method Settings

For all problems, a three(two)-layered feed-forward neural network was used for $f$ ($g$) in Eq. (1). For $f$, the number of output and hidden nodes was 100, and ReLU activation was used. For $h$ in Eq. (2), a three-layered feed-forward neural network with 100 hidden nodes with ReLU activation and 100 output nodes ($T = 100$) with the Softplus function was used. Hyperparameters were determined based on the empirical squared error for relative DRE on validation data. The dimension of latent vectors $\mathbf{z}$ was chosen from $\{4, 8, 16, 32, 64, 128, 256\}$. Relative parameter $\alpha$ was set to 0.5, which is a value recommended in a previous study [49]. We used the Adam optimizer [20] with a learning rate of 0.001. The mini-batch size was set to 256 (i.e., $N_{\mathcal{Q}} = 128$ for numerator and denominator instances). In training with source datasets, support instances are included in query instances as

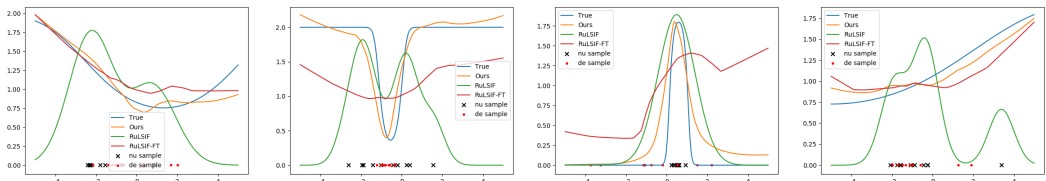

Figure 2: Illustrating examples of relative density-ratio estimation when 10 support instances are used in each target dataset. Horizontal and vertical axes represent $x$ and relative density-ratio values, respectively. Blue line denotes true relative density-ratio. Orange, green, and red lines represent estimated relative density-ratios by Ours, RuLSIF, and RuLSIF-FT, respectively.

in [9, 10]. The squared error on validation data was used for early stopping to avoid over-fitting, where the maximum number of training iterations was 10,000. This setup was used for all neural network-based methods in subsequent subsections. We implemented all methods by PyTorch [32].

## 5.2 Relative Density-ratio Estimation

We evaluate the relative DRE performance of the proposed method. We evaluated the squared error in Eq. (3) with $\alpha = 0.5$ on test instances ignoring the constant term that does not depend on models.

**Data** We used one synthetic data and two real-word benchmark data (Mnist-r[1] and Isolet[2]), which have been commonly used in transfer or multi-task learning studies [11, 27, 23]. In the synthetic data, each dataset $X_d$ was generated from a one-dimensional Gaussian distribution $\mathcal{N}(\mu_d, \sigma_d^2)$. Dataset-specific mean $\mu_d$ and standard deviation $\sigma_d$ were uniform randomly selected from $[-1.5, 1.5]$ and $[0.1, 2]$, respectively. Each dataset consists of 300 instances that are generated from each distribution. We created 600 source, 3 validation, 20 target datasets and evaluated the mean test squared error of all target dataset pairs when the number of target support instances was $N_{\mathcal{S}} = 10$. Mnist-r, which was derived from MNIST, consists of images. Mnist-r has six tasks, where each task is created by rotating the images in multiples of 15 degrees: 0, 15, 30, 45, 60, and 75. Each task has 1000 images, which are represented by 256-dimensional vectors, of 10 classes (digits). Isolet consists of letters spoken by 150 speakers, and speakers are grouped into five groups (tasks) by speaking similarity. Each instance is represented as a 617-dimensional vector. The number of classes (letters) is 26. For both benchmark data, we treat each class of each task as a dataset, and thus, Mnist-r and Isolet have 60 and 130 datasets, respectively. We randomly chose one task and then chose 10 target datasets from the task. From the remaining datasets, we randomly chose 10 validation sets and used the remaining as source datasets. We created 10 different splits of source/validation/target datasets and evaluated the mean test squared error of all target dataset pairs.

**Comparison methods** We compared the proposed method with RuLSIF [49] and RuLSIF-FT. RuLSIF trains a kernel model with only target support instances for relative DRE. We used the Gaussian kernel and Gaussian width was set to the median distance between support instances, which is a useful heuristic (median trick) [39]. Since neural network-based models performed poorly in our experiments due to small instances, we used the kernel model as in the original paper. RuLSIF-FT uses a neural network for modeling the relative density-ratio. RuLSIF-FT pretrains the model using source datasets and fine-tunes the weights of the last layer with target support instances. Note that the pretrained model in RuLSIF-FT cannot estimate the relative density-ratio for target datasets without fine-tuning. We used the same network architecture as the proposed method, i.e., the four-layered feed-forward neural network. For RuLSIF and RuLSIF-FT, regularization parameter $\lambda$ was chosen from $\{0.0001, 0.001, 0.01, 0.1, 1\}$, and the best test results were reported.

**Results** Figure 2 shows five illustrating examples of relative DRE in the synthetic data. The proposed method was able to accurately estimate the relative density-ratio from small target support instances. The mean test squared errors without the constant term of Ours, RuLSIF, and RuLSIF-FT were -0.613, -0.559, and -0.551, respectively (lower is better). Table 1 shows the mean test squared

---

[1] https://github.com/ghif/mtae    [2] http://archive.ics.uci.edu/ml/datasets/ISOLET

Table 1: Results for relative DRE: Average test squared errors ignoring the constant term with different target support instance sizes. Boldface denotes the best and comparable methods according to the paired t-test ($p = 0.05$).

| Data | $N_\mathcal{S}$ | Ours | RuLSIF | RuLSIF-FT |
|---|---|---|---|---|
| Mnist-r | 1 | **-0.671** | -0.543 | -0.503 |
| | 2 | **-0.748** | -0.581 | -0.513 |
| | 3 | **-0.772** | -0.600 | -0.518 |
| | 4 | **-0.784** | -0.597 | -0.520 |
| | 5 | **-0.793** | -0.578 | -0.520 |
| Avg. | | **-0.754** | -0.580 | -0.515 |
| Isolet | 1 | **-0.873** | -0.656 | -0.508 |
| | 2 | **-0.893** | -0.676 | -0.512 |
| | 3 | **-0.900** | -0.693 | -0.514 |
| | 4 | **-0.903** | -0.695 | -0.514 |
| | 5 | **-0.905** | -0.683 | -0.515 |
| Avg. | | **-0.895** | -0.681 | -0.513 |

Table 2: Results for dataset comparison: Average test AUCs [%] with different target support instance sizes. Boldface denotes the best and comparable methods according to the paired t-test ($p = 0.05$).

| Data | $N_\mathcal{S}$ | Ours | RuLSIF | uLSIF | D3RE | MMD | RuLSIF-FT | D3RE-FT |
|---|---|---|---|---|---|---|---|---|
| Mnist-r | 1 | **83.53** | 47.86 | 55.01 | 64.88 | 45.14 | 69.43 | 63.83 |
| | 2 | **93.00** | 84.46 | 78.09 | 73.26 | 85.09 | 78.28 | 68.29 |
| | 3 | **93.86** | 87.83 | 78.81 | 75.30 | 89.18 | 85.66 | 67.96 |
| | 4 | **96.49** | 93.90 | 83.51 | 80.19 | 93.08 | 87.31 | 71.79 |
| | 5 | 97.54 | **98.23** | 90.66 | 82.96 | **98.01** | 86.09 | 78.62 |
| Avg. | | **92.88** | 82.46 | 77.22 | 75.32 | 82.10 | 81.36 | 70.10 |
| Isolet | 1 | **96.28** | 50.18 | 59.38 | 81.48 | 48.11 | 79.50 | 81.50 |
| | 2 | **98.32** | 94.28 | 89.70 | 88.20 | 94.57 | 81.62 | 87.76 |
| | 3 | **99.23** | 97.22 | 94.27 | 89.69 | 97.79 | 85.93 | 88.39 |
| | 4 | **99.37** | 99.01 | 96.54 | 91.80 | **98.96** | 83.57 | 91.12 |
| | 5 | **99.60** | **99.61** | 96.77 | 93.86 | **99.43** | 83.19 | 92.74 |
| Avg. | | **98.56** | 88.07 | 87.33 | 89.00 | 87.77 | 82.76 | 88.30 |

Table 3: Ablation study for relative DRE. Average test squared errors without the constant term over different target support instance sizes.

| Data | Ours | No Latent | No Sadapt | NoSadapt-FT |
|---|---|---|---|---|
| Mnist-r | **-0.754** | -0.739 | -0.724 | -0.663 |
| Isolet | **-0.895** | -0.888 | -0.856 | -0.723 |
| Avg. | **-0.825** | -0.814 | -0.790 | -0.693 |

Table 4: Ablation study for dataset comparison. Average test AUCs [%] over different target support instance sizes.

| Data | Ours | No Latent | No Sadapt | NoSadapt-FT |
|---|---|---|---|---|
| Mnist-r | **92.88** | 91.66 | 88.46 | 89.92 |
| Isolet | **98.56** | 98.31 | 94.73 | 87.49 |
| Avg. | **95.72** | 94.99 | 91.60 | 88.71 |

errors ignoring the constant term with different target support instance sizes in Mnist-r and Isolet. The proposed method clearly outperformed RuLSIF and RuLSIF-FT. Since RuLSIF does not use source datasets, it performed worse than the proposed method. Although RuLSIF-FT uses source datasets, it did not work well since it does not have mechanisms for few-shot relative DRE. In contrast, the proposed method trains the model so that it explicitly improves test relative DRE performance after adapting to a few instances, and thus, it worked well.

Table 3 shows the results of an ablation study of the proposed method. NoLatent is our model without latent vectors for datasets $\mathbf{z}$. NoSadapt is our model without adapting to support instances with the closed-form solution $\hat{\mathbf{w}}$ in Eq. (5). NoSadapt learns dataset-invariant linear weights $\mathbf{w}$, and uses only latent vectors of target datasets $\mathbf{z}$ to estimate the relative density-ratio for the datasets. Thus, NoSadapt can be categorized into encode-decoder meta-learning methods. NoSadapt-FT finetunes the liner weights $\mathbf{w}$ in the model learned by NoSadapt with target support instances. Note that our model without both latent vectors and adapting to support instances cannot perform relative DRE for target datasets because it cannot take any information of target datasets. The details of these models are explained in the supplemental material. Although all methods performed better than RuLSIF and RuLSIF-FT, the proposed method outperformed the others. This result indicates that considering both latent vectors and adaptation to support instances is useful in our framework.

## 5.3 Dataset Comparison

We evaluate the proposed method with a dataset comparison problem. The aim of this problem is to determine if two datasets that consist of a few instances come from the same distribution. The proposed method outputs the score of whether two datasets come from the same distribution by calculating relative Pearson (PE) divergence [49], which is calculated using the relative density-ratio.

**Data**  We used Mnist-r and Isolet described in the previous subsection. We regard two datasets as coming from the same distribution if they are from the same class of the same task. We used all target dataset pairs for evaluation. Since the numbers of the same and different pairs in the target datasets are imbalanced (10 same and 90 different pairs), we used the area under ROC curve (AUC) as an evaluation metric because it can property evaluate the performance in imbalanced classification problems [2].

**Comparison methods**   We compared the proposed method with six methods: RuLSIF, uLSIF [17], deep direct DRE (D3RE) [18], maximum mean discrepancy (MMD) [12], RuLSIF-FT, and D3RE-FT. uLSIF is a DRE method, which is equivalent to RuLSIF with $\alpha = 0$. For RuLSIF, uLSIF, and RuLSIF-FT, the setting is the same as those of subsection 5.2. D3RE is a recently proposed neural network-based DRE method. We used the LSIF-based loss function and the same network architectures as the proposed method, i.e., the four-layered feed-forward neural network. D3RE-FT pretrains the model with source datasets and fine-tunes the weights of the last layer with target support instances. For D3RE and D3RE-FT, hyperparameter $C$ was chosen from $\{0.1, 0.5, 1, 10\}$, and the best test results were reported. MMD is a non-parametric distribution discrepancy metric, which is widely used since it can compare distributions without density estimation. We used the Gaussian kernel and Gaussian width was determined by the median trick. For the proposed method, RuLSIF, and RuLSIF-FT, relative PE divergence was used for the distribution discrepancy metric. For uLSIF, D3RE, and D3RE-FT, PE divergence was used. Although the proposed method, RuLSIF-FT, and D3RE-FT use source datasets for training, the others do not. Note that no methods use any information of similarity/dissimilarity of two datasets during training.

**Results**   Table 2 shows the mean test AUCs with different target support instance sizes. The proposed method showed the best/comparable results for all cases. RuLSIF performed better than uLSIF since relative DRE is more stable than DRE with a few instances. D3RE performed worse than the proposed method since target support instances were too small to train its neural network. When support instance size was small, the proposed method outperformed the others by a large margin. This is because it is difficult for RuLSIF, uLSIF, D3RE, and MMD to compare two datasets from only a few target instances. In contrast, the proposed method was able to accurately compare two datasets from a few instances because it learns to perform accurate relative DRE with a few instances. Since RuLSIF-FT and D3RE-FT were not trained for few-shot DRE, they did not perform well. Table 4 shows the results of the ablation study. Similar to the results in subsection 5.2, the proposed method performed better than the others by considering both latent vectors and adapting to support instances.

## 5.4   Inlier-based Outlier Detection

We evaluate the proposed method with an inlier-based outlier detection problem. This problem is to find outlier instances in an unlabeled dataset based on another dataset that consists of normal instances. By defining the density-ratio where the numerator and denominator densities are normal and unlabeled densities, respectively, we can see that the density-ratio values for outliers are close to zero since outliers are in a region where normal (unlabeled) density is low (high). Thus, we can use the negative density-ratio value as outlier scores [1, 14]. Similarly, the relative density-ratio values of outliers are close to zero, and thus, we can also use them as outlier scores [49]. In this problem, each dataset consists of normal and unlabeled instances $X_d = X_d^{\mathrm{nor}} \cup X_d^{\mathrm{un}}$. Along with this, we use a slightly modified sampling procedure of Algorithm 1. Specifically, for each iteration, we sample one dataset from the source datasets and create support and query instances from the dataset. The details of the algorithm are described in the supplemental material. We assume that the number of target normal support instances $N_S^{\mathrm{nor}}$ is small since labeling cost is often high in practice such as normal behavior-based outlier systems for new users [24].

**Data**   We used three real-world benchmark data: IoT[3], Landmine[4], and School[5]. These benchmark data are commonly used in outlier detection studies [23, 15]. IoT is real network traffic data, which are gathered from nine IoT devices (datasets) infected by malware. Landmine consists of 29 datasets, and each instance is extracted from a radar image that captures a region of a minefield. School contains the examination scores of students from 139 schools (datasets). We picked schools with 100 or more students, ending up with 74 datasets. The average outlier rates in a dataset of IoT, Landmine, and School are 0.05, 0.06, and 0.15, respectively. The details of the benchmark data are described in the supplemental material. For IoT, we randomly chose one target, one validation, and seven source datasets. For Landmine, we randomly chose 3 target, 3 validation, and 23 source datasets. For School, we randomly chose 10 target, 10 validation, and 54 source datasets. For each source/validation dataset in IoT, Landmine, and School, we chose 200, 200, and 50 instances, respectively, as normal and the remaining as unlabeled instances. For each target dataset, we used all instances except for

---

[3]  https://archive.ics.uci.edu/ml/datasets/detection_of_IoT_botnet_attacks_N_BaIoT    [4] http://people.ee.duke.edu/ lcarin/LandmineData.zip    [5] http://multilevel.ioe.ac.uk/intro/datasets.html

Table 5: Results for inlier-based outlier detection: Average test AUCs [%] with different target normal support instance sizes $N_S^{\mathrm{nor}}$. Boldface denotes the best and comparable methods according to the paired t-test ($p = 0.05$). Second column denotes the number of target normal support instances $N_S^{\mathrm{nor}}$.

| Data | | Ours | RuLSIF | uLSIF | D3RE | AE | SD | LOF | IF | AE-S | SD-S | AE-FT | SD-FT | RuLSIF-FT | D3RE-FT |
|------|---|------|--------|-------|------|-----|-----|-----|-----|------|------|-------|-------|-----------|---------|
| IoT | 1 | **97.28** | **95.75** | **95.87** | 85.20 | 93.09 | 91.90 | 93.30 | 41.32 | 43.59 | 40.12 | 66.05 | 55.50 | **93.15** | 84.51 |
| | 2 | **97.81** | 93.96 | 94.09 | 87.48 | 92.41 | 92.20 | 93.29 | 45.84 | 43.60 | 34.63 | 75.21 | 67.65 | 88.53 | 89.57 |
| | 3 | **96.39** | 94.64 | **95.38** | 84.33 | **89.48** | 90.86 | **93.34** | 40.79 | 43.61 | 36.57 | 76.88 | 72.82 | **89.38** | 88.97 |
| | 4 | **97.05** | **94.46** | 93.43 | 81.74 | **90.75** | 89.76 | 92.82 | 42.66 | 43.60 | 40.26 | 80.53 | 67.27 | **89.01** | 89.45 |
| | 5 | **96.17** | **94.43** | **95.29** | 81.49 | **89.01** | 87.49 | 92.87 | 40.14 | 43.65 | 38.67 | 80.25 | 73.56 | **88.96** | 89.12 |
| Avg. | | **96.94** | 94.65 | 94.81 | 84.05 | 90.95 | 90.44 | 93.12 | 42.15 | 43.61 | 38.05 | 75.78 | 67.36 | 89.80 | 88.33 |
| Land mine | 1 | **68.70** | 53.69 | 53.84 | 49.54 | 52.91 | 52.60 | 45.09 | 55.80 | 52.79 | 50.36 | 55.49 | 53.95 | 60.86 | 52.15 |
| | 2 | **64.92** | **55.56** | **55.21** | 52.54 | 50.39 | 53.00 | 45.17 | **56.08** | 52.79 | 51.16 | 52.08 | 52.35 | **62.55** | 55.15 |
| | 3 | **63.66** | **54.74** | 54.41 | 51.84 | 49.50 | 49.16 | 45.17 | **56.90** | 52.80 | 50.97 | **55.43** | 53.67 | **61.36** | 53.34 |
| | 4 | **66.24** | 54.94 | 53.98 | 52.21 | 49.65 | 51.74 | 45.12 | 55.73 | 52.80 | 50.08 | 55.40 | 52.40 | 62.04 | 51.90 |
| | 5 | **63.05** | **55.46** | 53.42 | 53.32 | 50.88 | 51.08 | 45.19 | **56.35** | 52.79 | 50.08 | **54.43** | **54.34** | **62.62** | 54.46 |
| Avg. | | **65.31** | 54.88 | 54.17 | 51.89 | 50.67 | 51.52 | 45.15 | 56.17 | 52.80 | 50.53 | 54.37 | 53.34 | 61.89 | 53.40 |
| Sch ool | 1 | **62.98** | 55.00 | 54.99 | 53.05 | 56.27 | 54.63 | 53.94 | 57.44 | 58.32 | 56.36 | **59.07** | 56.26 | 56.26 | 52.46 |
| | 2 | **62.18** | 56.34 | 54.81 | 53.79 | 57.20 | 56.24 | 53.73 | 57.11 | 58.27 | 56.64 | **59.27** | **58.53** | 56.02 | 53.47 |
| | 3 | **64.30** | 56.69 | 55.93 | 54.81 | 57.54 | 56.71 | 54.26 | 58.28 | 58.26 | 57.10 | 59.51 | 57.25 | 55.36 | 54.28 |
| | 4 | **63.70** | 58.15 | 57.42 | 54.78 | 58.46 | 57.63 | 54.10 | 57.09 | 58.25 | 56.15 | 59.70 | 55.66 | 55.60 | 55.61 |
| | 5 | **64.61** | 57.76 | 57.71 | 54.89 | 58.54 | 57.01 | 54.12 | 56.92 | 58.24 | 56.75 | 59.33 | 56.84 | 56.11 | 56.02 |
| Avg. | | **63.55** | 56.79 | 56.17 | 54.26 | 57.60 | 56.45 | 54.03 | 57.13 | 58.27 | 56.48 | 59.38 | 56.91 | 55.87 | 54.37 |

target normal support instances as unlabeled instances (test instances). For each benchmark data, we randomly created 10 different splits of target/validation/source datasets and evaluated the mean test AUC on the target datasets.

**Comparison methods**   We compare the proposed method with 13 outlier detection methods: RuL-SIF, uLSIF [14], D3RE, local outlier factor (LOF) [5], isolation forest (IF) [28], autoencoder (AE) [38], deep support vector description (SD) [36], AE-S, SD-S, fine-tuning methods for AE and SD (AE-FT and SD-FT), RuLSIF-FT, and D3RE-FT. LOF and IF use only target unlabeled instances to find outliers. AE and SD use target normal instances for training. AE-S and SD-S use source normal instances for training. AE-FT and SD-FT fine-tune models trained by AE-S and SD-S with target normal instances, respectively. Note that although AE and SD-based methods can use unlabeled instances as well as normal instances for training, they performed worse than them trained with only normal instances. Thus, we used only normal instances for training. RuLSIF, uLSIF, and D3RE use target normal and unlabeled instances. RuLSIF-FT and D3RE-FT use source normal and unlabeled instances as well as target normal and unlabeled instances. Note that no methods use any information of outliers for training. The details of the implementation such as neural network architectures and hyperparameter candidates are described in the supplemental material.

**Results**   Table 5 shows the mean test AUCs with different target normal support instance sizes. The proposed method showed the best/comparable results for all cases. Density-ratio methods such as RuLSIF and uLSIF tended to show better results than the other comparison methods by using information from both target normal and unlabeled instances. The proposed method was able to further improve performance than RuL-SIF and uLSIF by incorporating the mechanism for few-shot relative DRE. Table 6 shows the results of an

Table 6: Ablation study for outlier detection. Average test AUCs [%] over different target normal support instance sizes.

| Data | Ours | No Latent | No Sadapt | NoSadapt -FT |
|------|------|-----------|-----------|--------------|
| IoT | 96.94 | **97.63** | 95.17 | 94.44 |
| Landmine | 65.31 | 63.08 | **68.83** | 65.83 |
| School | **63.55** | 63.40 | 60.04 | 62.21 |
| Avg. | **75.27** | 74.70 | 74.49 | 74.16 |

ablation study. The best method can vary across benchmark data since each benchmark data has different properties. For example, in IoT, all datasets are relatively similar, which is validated by the fact that test AUCs are high [23]. Thus, the dataset-invariant embedding function $h$ in NoLatent is sufficient for adaptation. Nevertheless, the proposed method (Ours) showed the best average AUCs over all benchmark data. In the supplemental material, we additionally showed that the proposed method outperformed the PU learning method [21].

## 5.5   Dependency of Relative Parameter $\alpha$

We investigated the dependency of relative parameter $\alpha > 0$ in the proposed method. Relative parameter $\alpha$ determines the upper bound of relative density-ratio value since $r_\alpha(\mathbf{x}) \leq \frac{1}{\alpha}$ for any

Table 7: Investigation of dependency of relative parameter $\alpha$ in the proposed method. Values in relative DRE represent average test squared errors ignoring constant terms over different target support instance sizes and all benchmark data. Values in dataset comparison and outlier detection represent average test AUCs [%] over different target support instance sizes and all benchmark data.

| Problem | Ours $\alpha=0.1$ | Ours $\alpha=0.5$ | Ours $\alpha=0.9$ | RuLSIF $\alpha=0.1$ | RuLSIF $\alpha=0.5$ | RuLSIF $\alpha=0.9$ |
|---|---|---|---|---|---|---|
| relative DRE | -2.80 | -0.83 | -0.53 | -0.64 | -0.63 | -0.48 |
| dataset comparison | 92.81 | 95.72 | 95.03 | 85.51 | 85.27 | 86.34 |
| outlier detection | 74.77 | 75.27 | 74.98 | 68.47 | 68.77 | 68.69 |

Table 8: Computation time in seconds for each method on dataset comparison. Ours (train) and RuLSIF-FT (train) represent training time with source datasets for Ours and RuLSIF-FT, respectively. Ours (test), RuLSIF-FT (test), RuLSIF, uLSIF, and MMD represent test time for 100 target dataset comparisons.

| Ours (train) | Ours (test) | RuLSIF | uLSIF | MMD | RuLSIF-FT (train) | RuLSIF-FT (test) |
|---|---|---|---|---|---|---|
| 137.93 | 0.32 | 0.38 | 0.34 | 0.12 | 50.32 | 0.24 |

**x.** Table 7 shows results with $\alpha = 0.1, 0.5$, and $0.9$ for all three problems. The proposed method consistently outperformed RuLSIF over different $\alpha$ values. This result suggests that the proposed method is relatively robust against the relative parameter value. Note that, in relative DRE, comparison between different $\alpha$ values is meaningless since the scale of the evaluation metrics is different. Besides, various additional results such as investigation of the dependency of the dimensions of latent vectors are described in the supplemental material.

### 5.6 Computation Cost

We investigated the computation time of the proposed method. Table 8 shows the computation time of each method for dataset comparison with Mnist-r. We used a computer with a 2.20GHz CPU. The support instance size in each target dataset was set to five. We omitted D3RE since it requires to train the neural network for each target dataset comparison, which is quite time-consuming compared to the others. Although the proposed method took time for training with source datasets, it was able to compare datasets with relative DRE as fast as other methods.

## 6 Limitations

The proposed method uses multiple source datasets to improve relative DRE performance on target datasets. However, when source and target datasets are significantly different, there is a risk of degrading the performance on the target datasets. This phenomenon is called "negative transfer", and is a common challenge in general transfer/meta-learning methods. Developing methods to automatically remove negative effects of such datasets is one of the important research directions.

## 7 Conclusion

In this paper, we proposed a meta-learning method for relative DRE. We empirically showed that the proposed method outperformed various existing methods in three problems: relative DRE, dataset comparison, and outlier detection. As future work, we plan to incorporate other DRE models such as telescopic DRE [35] in our framework. We describe a potential negative social impact of our work. The proposed method needs to access datasets obtained from multiple sources like almost all meta-learning methods. When each dataset is provided from different owners, sensitive information in the dataset risks being stolen and abused by malicious people. To evade this risk, we encourage research for developing meta-learning methods without accessing raw datasets.

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
