# Supplemental Material:
# Meta-Learning for Relative Density-Ratio Estimation

**Atsutoshi Kumagai**
NTT Computer and Data Science Laboratories
atsutoshi.kumagai.ht@hco.ntt.co.jp

**Tomoharu Iwata**
NTT Communication Science Laboratories
tomoharu.iwata.gy@hco.ntt.co.jp

**Yasuhiro Fujiwara**
NTT Communication Science Laboratories
yasuhiro.fujiwara.kh@hco.ntt.co.jp

## 1  Pseudocode of the Proposed Method for Inlier-based Outlier Detection

The pseudocode of the proposed method for inlier-based outlier detection is illustrated in Algorithm 1. We slightly modify the sampling procedure (Lines 2–5) for inlier-based outlier detection. For each iteration, we randomly sample one dataset that consists of normal and unlabeled instances (Line 2). From the dataset, we randomly select normal and unlabeled support instances $\mathcal{S} = \mathcal{S}_{\mathrm{nor}} \cup \mathcal{S}_{\mathrm{un}}$ and query instances $\mathcal{Q} = \mathcal{Q}_{\mathrm{nu}} \cup \mathcal{Q}_{\mathrm{de}}$ (Lines 3–5). We then calculate the relative density-ratio with support instances (Line 6). Using the learned relative density-ratio, we calculate loss $\tilde{J}_\alpha(\mathcal{Q}; \mathcal{S})$ with the query instances (Line 7). Lastly, the parameters of our model are updated with the gradient of the loss (Line 8). In the inlier-based outlier detection experiments, the number of unlabeled support instances $N_{\mathcal{S}}^{\mathrm{un}}$ was set to 100 during training with source datasets.

## 2  Details of Benchmark Data for Inlier-based Outlier Detection

We used three real-world benchmark data: IoT[1], Landmine[2], and School[3]. These benchmark data are commonly used in outlier detection studies [14, 9].

IoT is real network traffic data, which are gathered from nine IoT devices (datasets) infected by malware. Each dataset has normal and malicious traffic instances, and each instance is represented by a 115-dimensional vector. For each dataset, we randomly used 475 normal and 25 malicious (outlier) instances. We normalized each feature vector by $\ell_2$-normalization.

Landmine consists of 29 datasets, and each instance is a nine-dimensional vector extracted from a radar image that captures a region of a minefield. Each dataset has 513 instances on average. We normalized each feature vector by $\ell_2$-normalization.

School contains the examination scores of students from 139 schools (datasets). We picked schools with 100 or more students, ending up with 74 datasets. We used binary features only resulting in $M = 26$. We created outlier/normal instances on the basis of whether the examination score was 35 or more. The average outlier rates in a dataset of IoT, Landmine, and School are 0.05, 0.06, and 0.15, respectively.

For IoT, we randomly chose one target, one validation, and seven source datasets. For Landmine, we randomly chose 3 target, 3 validation, and 23 source datasets. For School, we randomly chose 10 target, 10 validation, and 54 source datasets. For each source/validation dataset in IoT, Landmine, and

---

[1] https://archive.ics.uci.edu/ml/datasets/detection_of_IoT_botnet_attacks_N_BaIoT
[2] http://people.ee.duke.edu/ lcarin/LandmineData.zip
[3] http://multilevel.ioe.ac.uk/intro/datasets.html

35th Conference on Neural Information Processing Systems (NeurIPS 2021).

---
**Algorithm 1** Training procedure of our model for inlier-based outlier detection.
---
**Require:** Datasets $X = \{X_d\}_{d=1}^D = \{X_d^{\text{nor}} \cup X_d^{\text{un}}\}_{d=1}^D$, normal support instance size $N_\mathcal{S}^{\text{nor}}$,
    unlabeled support instance size $N_\mathcal{S}^{\text{un}}$, query instance size $N_\mathcal{Q}$, relative parameter $\alpha$
**Ensure:** Parameters of our model $\Theta$
 1: **repeat**
 2:    Sample one dataset $d$ from $\{1, \ldots, D\}$
 3:    Select normal support instances $\mathcal{S}_{\text{nor}}$ with size $N_\mathcal{S}^{\text{nor}}$ from $X_d^{\text{nor}}$
 4:    Select unlabeled support instances $\mathcal{S}_{\text{un}}$ with size $N_\mathcal{S}^{\text{un}}$ from $X_d^{\text{un}}$
 5:    Select query instances $\mathcal{Q}_{\text{nu}}$ and $\mathcal{Q}_{\text{de}}$ with size $N_\mathcal{Q}$ from $X_d^{\text{nor}}$ and $X_d^{\text{un}}$, respectively
 6:    Calculate linear weights $\tilde{\mathbf{w}}$ with the support sets by Eq. (5) to obtain the relative density-ratio
       Eq. (6)
 7:    Calculate the loss $\tilde{J}_\alpha(\mathcal{Q}; \mathcal{S})$ in Eq. (8) with the query instances
 8:    Update parameters with the gradients of the loss $\tilde{J}_\alpha(\mathcal{Q}; \mathcal{S})$
 9: **until** End condition is satisfied;
---

School, we chose 200, 200, and 50 instances, respectively, as normal and the remaining as unlabeled instances. For each target dataset, we used all instances except for target normal support instances as unlabeled instances (test instances). For each benchmark data, we randomly created 10 different splits of target/validation/source datasets and evaluated the mean test AUC on the target datasets.

## 3 Details of Comparison Methods for Inlier-based Outlier Detection

We compare the proposed method with 13 outlier detection methods: RuLSIF, uLSIF [7], D3RE [11], local outlier factor (LOF) [2], isolation forest (IF) [15], autoencoder (AE) [19], deep support vector description (SD) [18], AE-S, SD-S, fine-tuning methods for AE and SD (AE-FT and SD-FT), RuLSIF-FT, and D3RE-FT.

LOF and IF use only target unlabeled instances to find outliers. For LOF, the number of neighbors was chosen from $\{10, 20, 30, 40, 50\}$. For IF, the number of base estimators was chosen from $\{10, 50, 100, 200\}$. For both methods, the best test AUCs were reported.

AE and SD use target normal instances for training. AE-S and SD-S use source normal instances for training. For AE and AE-S, we used two four-layered feed-forward networks for the encoder and decoder, respectively. The output dimension of the encoder was chosen from $\{4, 8, 16, 32, 64, 128, 256\}$. For SD and SD-S, we used the same neural network architectures as the proposed method, i.e., four-layered feed-forward neural networks. The output dimension of neural networks was chosen from $\{4, 8, 16, 32, 64, 128, 256\}$. For AE, AE-S, SD, and SD-S, these hyperparameters were determined on the basis of outlier scores on validation normal data. AE-FT and SD-FT fine-tune models trained by AE-S and SD-S with target normal instances, respectively. For both methods, the number of iterations for fine-tuning was chosen from $\{1, 5, 10, 20, 30, 40, 50, 100, 200, 300, 400, 500\}$, and the best test AUCs were reported. Note that although AE and SD-based methods can use unlabeled instances as well as normal instances for training, they performed worse than them trained with only normal instances. Thus, we used only normal instances for training.

RuLSIF, uLSIF, and D3RE use target normal and unlabeled instances. RuLSIF-FT and D3RE-FT use source normal and unlabeled instances as well as target normal and unlabeled instances. Note that no methods use any information of outliers for training.

## 4 Intuitive Explanation of Why the Proposed Method Works Well

The meta-learning methods including the proposed method learn how to learn from small data using various datasets [8, 24]. Specifically, they directly maximize the expected test performance of a model after it has been trained on small data by using various datasets. To retain the knowledge of how to learn, representative meta-learning methods such as MAML [3], neural processes [4, 5], and prototypical networks [21] use shared neural networks across datasets. The proposed method also use shared neural networks $f$, $g$, and $h$. Although each dataset has potentially small instances, the total number of instances over all datasets would be sufficiently large to train such neural networks

without overfitting. The high expressive capabilities of neural networks enable the model to store the knowledge. In fact, many studies have reported that these methods empirically work well [3, 4, 5, 21, 17, 10]. This is an intuitive explanation of the meta-learning methods including the proposed method.

# 5 Details of the Variants of Our Model

We explain the variants of our model (NoLatent, NoSadapt, and NoSadapt-FT) used in the ablation studies in detail.

**NoLatent** is our model without latent vectors for datasets $\mathbf{z}$. Specifically, this method uses $\hat{r}_\alpha(\mathbf{x}) := \mathbf{w}^\top h(\mathbf{x})$ as the relative DRE model. Although the parameters of neural network $h$ are dataset-invariant, datasets-specific linear weights $\mathbf{w}$ can be obtained with support instances by Eq. (5) while fixing $h$. Therefore, NoLatent can perform relative DRE by using the target dataset (support instances) information.

**NoSadapt** is our model without adapting to support instances with the closed-form solution $\hat{\mathbf{w}}$ in Eq. (5). This method uses $\hat{r}_\alpha(\mathbf{x}) := \mathbf{w}^\top h([\mathbf{x}, \mathbf{z}_{\mathrm{nu}}, \mathbf{z}_{\mathrm{de}}])$ as the relative DRE model. Here, both linear weights $\mathbf{w}$ and neural network parameters of $h$ are dataset-invariant, i.e., they are usual trainable parameters and are trained with source datasets. Latent vectors $\mathbf{z}_{\mathrm{nu}}$ and $\mathbf{z}_{\mathrm{de}}$ are used to treat support instance information. NoSadapt performs relative DRE on target datasets only by inferring $\mathbf{z}_{\mathrm{nu}}$ and $\mathbf{z}_{\mathrm{de}}$ from the target support instances. However, it might be difficult to perform accurate relative DRE since it does not explicitly minimize the squared error of relative DRE on the target support instances Eq. (3).

**NoSadapt-FT** further fine-tunes linear weights $\mathbf{w}$ of the learned model of NoSadapt by minimizing the squared error of relative DRE on the target support instances to improve the estimation performance. As a result, we obtain Eq. (5). However, it might be also difficult to improve the performance since the learned model with source datasets is not designed to be fine-tuned.

**Our method** directly improves the expected test performance of the fine-tuning with support instances using the meta-learning framework. Specifically, the fine-tuned model with support instances (Eq. 6) is evaluated on query instances (Eq. 8) in source datasets, and its loss can be backpropagated with standard gradient descent methods to improve the expected test performance. This can be performed since closed-form solution $\hat{\mathbf{w}}$ is differentiable w.r.t. the neural network parameters. Our method performs relative DRE on target datasets by both estimating latent vectors of datasets and adapting linear weights to the target datasets. For another perspective, we can regard $\hat{\mathbf{w}}$ as a neural network layer, and all the neural networks including this special differentiable layer are trained by minimizing the loss on query instances (Eq. 8) as standard neural network training.

# 6 Other Applications

In this section, we explain several potential applications of the proposed method.

## 6.1 Covariate Shift Adaptation

We consider a situation where training and test data follow different feature distributions but the label distribution given the features is constant, which is called "covariate shift" [20]. To learn a model that fits on the test distribution, (relative) density-ratio of the training and test feature distributions can be used for reweighting labeled training instances [20]. By learning our model with various source datasets, our model can accurately estimate the relative density-ratio of new training and test feature distributions even if a few instances that follow the new distributions are available.

## 6.2 Change Point Detection

The objective of change-point detection is to find abrupt changes lying behind time-series data. By comparing the probability distributions of subsequences of the time-series over past and present

Table 1: Comparison between relative DRE and DRE for dataset comparison. Average test AUCs [%] over different target support instance sizes.

| Data | Ours | Ours for DRE |
|---|---|---|
| Mnist-r | 92.88 | 60.80 |
| Isolet | 98.56 | 74.79 |
| Avg. | 95.72 | 67.80 |

Table 2: Investigation of effects of trainable relative parameter. Values in relative DRE represent average test squared errors ignoring constant terms over different target support instance sizes and all benchmark data. Values in dataset comparison and outlier detection represent average test AUCs [%] over different target support instance sizes and all benchmark data. Our w/ train represents our method with trainable relative parameter $\alpha$ for support set adaptation.

| Problem | Ours | Ours w/ train |
|---|---|---|
| relative DRE | -0.83 | -0.83 |
| dataset comparison | 95.72 | 95.78 |
| outlier detection | 75.27 | 75.45 |

intervals, we can detect the changes. Relative PE divergence can be used for measuring the two probability distributions, which is calculated based on the relative density-ratio [16]. By using information in various source datasets, the proposed method can accurately perform change-point detection even if a few subsequences are available in the past and/or present intervals.

### 6.3 Generative Adversarial Networks

Generative adversarial networks (GANs) are powerful generative models that consist of two components: a generator and discriminator [6]. The generator produces instances with the same distribution as training data and the discriminator aims to distinguish instances came from the training data and that from the generator. GANs are formulated as a two-player minmax game, and the generator and discriminator are optimized alternately. Training of the discriminator can be formulated as (relative) DRE between the training data distribution and generated distribution and training of the generator can be performed with the learned (relative) density-ratio [23]. Thus, by meta-learning with various related datasets in advance, the proposed method might improve the performance of GANs even if only a few training instances are available.

## 7 Additional Experimental Results

### 7.1 Comparison between Relative DRE and DRE in Our Framework

We investigated the DRE performance of the proposed method, i.e., the proposed method with $\alpha = 0$, with the dataset comparison problem. Table 1 shows the mean test AUCs over different target support instance sizes. Ours, which makes relative DRE with $\alpha = 0.5$, clearly performed better than Ours for DRE. Since neural networks are flexible, our model for DRE tried to fit on extremely large values of density-ratio, which led to the inaccurate performance. This result suggests that relative DRE is useful in our framework.

### 7.2 Trainable Relative Parameter $\alpha$ for Support Set Adaptation

We investigated the performance of the proposed method that uses trainable relative parameter $\alpha$ for adaptation to support instances. Although the proposed method in the main paper uses the same relative parameter (hyperparameter) in Eqs. (3) and (8) for simplicity, we can use different relative parameters and treat the relative parameter in Eq. (3) as a trainable parameter. Table 2 shows the mean test squared errors without the constant term for the relative DRE problems and the mean test AUCs for the dataset comparison and outlier detection problems. We fixed $\alpha = 0.5$ in Eq. (8) for both methods. We can see that although Ours w/ train tended to perform slightly better than Ours (the proposed method without the trainable relative parameter), there are no big differences between them. Therefore, in practice, we can use both methods.

Table 3: Investigation of effects of using deeper neural networks in the proposed method. Average test squared errors for relative DRE without the constant term for the relative DRE problems and average test AUCs [%] for the dataset comparison and outlier detection problems. DeepOurs uses a deeper neural network for $h$ than Ours.

| Problem | Ours | DeepOurs | RuLSIF |
|---|---|---|---|
| relative DRE | -0.83 | -0.78 | -0.63 |
| dataset comparison | 95.72 | 91.20 | 85.27 |
| outlier detection | 75.27 | 72.40 | 68.77 |

Table 4: Additional comparisons with several DRE methods. Values in relative DRE represent average test squared errors ignoring constant terms over different target support instance sizes and all benchmark data. Values in dataset comparison and outlier detection represent average test AUCs [%] over different target support instance sizes and all benchmark data.

| Problem | Data | Ours | PC | KNN | LRuSIF | PC-FT |
|---|---|---|---|---|---|---|
| relative DRE | Mnist-r | -0.754 | -0.725 | -0.633 | -0.610 | -0.565 |
| | Isolet | -0.895 | -0.873 | -0.776 | -0.613 | -0.568 |
| dataset comparison | Mnist-r | 92.88 | 86.22 | 80.08 | 84.35 | 78.81 |
| | Isolet | 98.56 | 97.06 | 86.51 | 45.20 | 82.66 |
| outlier detection | IoT | 96.94 | 76.72 | 50.72 | 89.37 | 85.05 |
| | Landmine | 65.31 | 50.47 | 50.19 | 42.17 | 53.19 |
| | School | 63.55 | 54.29 | 52.86 | 53.73 | 54.38 |

Table 5: Additional comparisons in the outlier detection problem. Average test AUCs [%] over different target support instance sizes.

| Data | Ours | KL | nnPU |
|---|---|---|---|
| IoT | 96.94 | 94.79 | 91.18 |
| Landmine | 65.31 | 54.80 | 51.60 |
| School | 63.55 | 58.27 | 54.15 |

## 7.3 Dependency of the Dimension of Latent Vectors

We investigated the dependency of the dimension of latent vectors $\mathbf{z} \in \mathbb{R}^K$ in the proposed method. The latent vector $\mathbf{z}$ is used for reflecting datasets' information to the embeddings of instances in the datasets. Figure 1 shows the average test squared errors without the constant term when varying $K$ on the relative DRE problems. Figures 2 and 3 show the average test AUCs when varying $L$ in the dataset comparison and outlier detection problems, respectively. The proposed method constantly outperformed RuLSIF over different $K$ in all problems. These results show that the proposed method is relatively robust against the dimension of latent vectors.

## 7.4 Deep Neural Network-based Implementation of the Proposed Method

We investigated the effect of using deeper neural networks in the proposed method. Table 3 shows the results. DeepOurs uses a five-layered feed-forward neural network for $h$ although Ours uses a three-layered one, which is used in the main paper. Although DeepOurs consistently outperformed RuLSIF, it performed worse than Ours for all problems. This result suggests that the neural network used in the main paper is more appropriate for our problems.

## 7.5 Additional Comparison with Density-Ratio Estimation Methods

We additionally compared the proposed method with four DRE methods: the probabilistic classification-based method with a logistic regression model (PC) [1], the nearest neighbour-based method (KNN) [13], RuLSIF with a liner model $\hat{r}(\mathbf{x}) := \mathbf{w}^\top \mathbf{x}$ (LRuLSIF), and a fine-tuning method for PC (PC-FT) that uses a neural network model and fine-tunes the last layer of the learned model of PC with source datasets by using target support instances. PC, KNN, and LRuLSIF use only target support instances. We note that a fine-tuning method for LRuLSIF was omitted since it is equivalent to RuLSIF-FT in the main paper. Since these methods use simple DRE models, they are considered

to be suitable for small instance problems. Therefore, we conducted this additional comparison. Table 4 shows mean test AUCs over different target support sample sizes. The proposed method clearly outperformed the other methods on all problems. This is because the proposed method is well designed for few-shot relative DRE.

### 7.6 Additional Comparison for Inlier-based Outlier Detection

To further demonstrate the effectiveness of the proposed method, we additionally compared the proposed method with two comparison methods: DRE method with unconstrained KL divergence (KL) [22] and non-negative positive and unlabeled (PU) learning method (nnPU) [12]. KL used a kernel model with the Gaussian kernel, where Gaussian width was calculated using the median trick. nnPU is a neural network-based PU learning method that learns classifiers from only positive and unlabeled data. This method is designed to mitigate overfitting to small data, and can be used for inlier-based outlier detection by regarding normal data as positive data. We used the same neural network architecture as the proposed method, i.e., the four-layered feed-forward neural network. Since nnPU requires the class-prior probability as domain knowledge, we set it by calculating the average outlier rate in a dataset for each benchmark data. We fixed hyperparameters $\beta = 0$ and $\gamma = 1$ as the original paper [12]. Both methods use target normal and unlabeled instances for training. Table 5 shows mean test AUCs over different target support sample sizes. The proposed method clearly outperformed the others. nnPU did not perform well since there are too small normal instances to train its neural network.

### 7.7 RuLSIF with Meta-Learned Initial Parameters

To further investigate the effectiveness of our formulation, we considered a meta-learning extension of RuLSIF, in which initial parameters of the kernel model in RuLSIF are meta-learned with source datasets like MAML [3]. We call this method Meta-RuLSIF. We conducted relative DRE experiments with the synthetic data used in the main paper. The mean squared errors without the constant terms of the proposed method and Meta-RuLSIF were -0.613 and -0.498, respectively (lower is better). The reason for the bad result of Meta-RuLSIF is that it has less expressive capability than our neural network-based embedding extractors and it learns the single initialization for all datasets even though good initialization would be different across datasets.

## 8 Potential Negative Social Impacts

The proposed method can be used for various real-world applications as described in Section 1. However, the proposed method has some potential risks to be addressed when it is deployed to real-world applications. First, although we experimentally demonstrated the proposed method outperformed existing methods, it is not perfect; that is, it may estimate incorrect relative density-ratio, which may lead to wrong decision making in some cases. To mitigate this, people can use the proposed method as a support tool for their decision making. Second, the proposed method needs to access datasets obtained from multiple sources like almost all transfer/meta-learning methods. When each dataset is provided from different owners such as companies, sensitive information in the dataset risks being stolen and abused by malicious people that use the proposed method. To evade this risk, we suggest promoting research for developing meta-learning methods without accessing raw datasets.

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

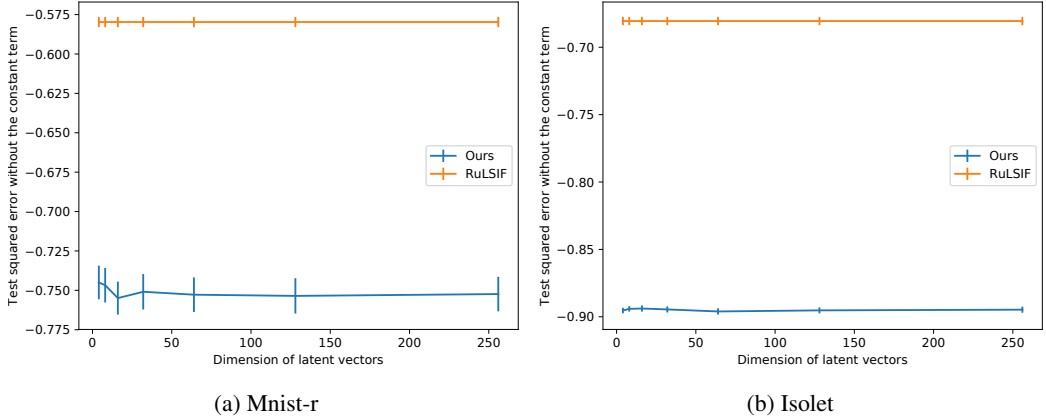

|            |            |
|:----------:|:----------:|
| (a) Mnist-r | (b) Isolet |

Figure 1: Average and standard errors of test squared errors without the constant term when varying the number of dimension of latent vectors $K$ on relative-density ratio estimation problems.

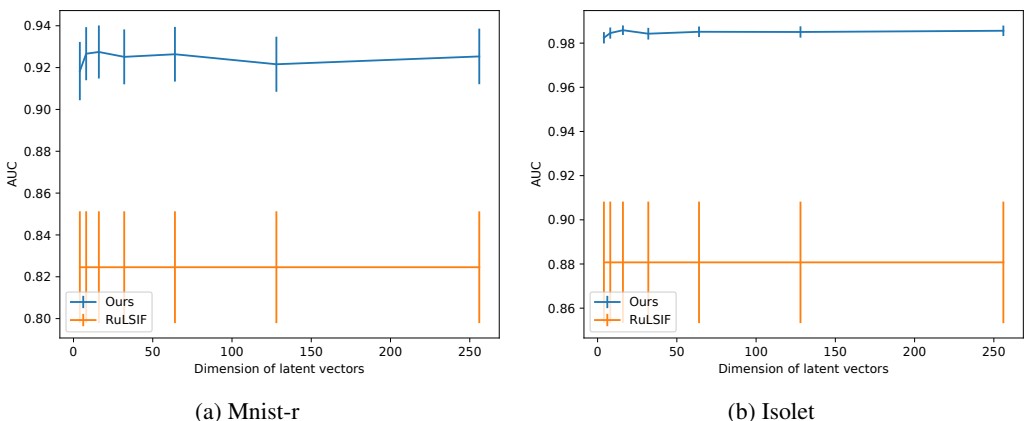

|            |            |
|:----------:|:----------:|
| (a) Mnist-r | (b) Isolet |

Figure 2: Average and standard errors of test AUCs when varying the number of dimension of latent vectors $K$ on dataset comparison problems.

[5] M. Garnelo, J. Schwarz, D. Rosenbaum, F. Viola, D. J. Rezende, S. Eslami, and Y. W. Teh. Neural processes. *arXiv preprint arXiv:1807.01622*, 2018.

[6] I. Goodfellow, J. Pouget-Abadie, M. Mirza, B. Xu, D. Warde-Farley, S. Ozair, A. Courville, and Y. Bengio. Generative adversarial nets. In *NeurIPS*, 2014.

[7] S. Hido, Y. Tsuboi, H. Kashima, M. Sugiyama, and T. Kanamori. Statistical outlier detection using direct density ratio estimation. *Knowledge and information systems*, 26(2):309–336, 2011.

[8] T. Hospedales, A. Antoniou, P. Micaelli, and A. Storkey. Meta-learning in neural networks: a survey. *arXiv preprint arXiv:2004.05439*, 2020.

[9] T. Idé, D. T. Phan, and J. Kalagnanam. Multi-task multi-modal models for collective anomaly detection. In *ICDM*, 2017.

[10] T. Iwata and A. Kumagai. Meta-learning from tasks with heterogeneous attribute spaces. In *NeurIPS*, 2020.

[11] M. Kato and T. Teshima. Non-negative bregman divergence minimization for deep direct density ratio estimation. In *ICML*, 2021.

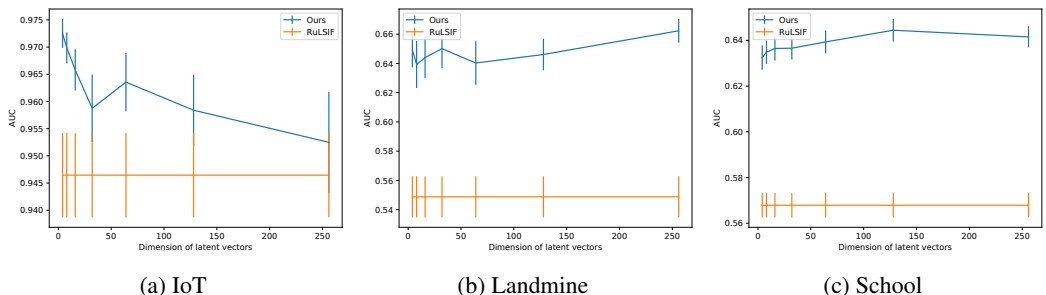

|     |     |     |
| :-: | :-: | :-: |
| (a) IoT | (b) Landmine | (c) School |

Figure 3: Average and standard errors of test AUCs when varying the number of dimension of latent vectors $K$ on inlier-based outlier detection problems.

[12] R. Kiryo, G. Niu, M. C. d. Plessis, and M. Sugiyama. Positive-unlabeled learning with non-negative risk estimator. In *NeurIPS*, 2017.

[13] J. Kremer, F. Gieseke, K. S. Pedersen, and C. Igel. Nearest neighbor density ratio estimation for large-scale applications in astronomy. *Astronomy and Computing*, 12:67–72, 2015.

[14] A. Kumagai, T. Iwata, and Y. Fujiwara. Transfer anomaly detection by inferring latent domain representations. In *NeurIPS*, 2019.

[15] F. T. Liu, K. M. Ting, and Z.-H. Zhou. Isolation forest. In *ICDM*, 2008.

[16] S. Liu, M. Yamada, N. Collier, and M. Sugiyama. Change-point detection in time-series data by relative density-ratio estimation. *Neural Networks*, 43:72–83, 2013.

[17] A. Rajeswaran, C. Finn, S. M. Kakade, and S. Levine. Meta-learning with implicit gradients. In *NeurIPS*, 2019.

[18] L. Ruff, N. Görnitz, L. Deecke, S. A. Siddiqui, R. Vandermeulen, A. Binder, E. Müller, and M. Kloft. Deep one-class classification. In *ICML*, 2018.

[19] M. Sakurada and T. Yairi. Anomaly detection using autoencoders with nonlinear dimensionality reduction. In *Proceedings of the MLSDA 2014 2nd Workshop on Machine Learning for Sensory Data Analysis*, page 4. ACM, 2014.

[20] H. Shimodaira. Improving predictive inference under covariate shift by weighting the log-likelihood function. *Journal of statistical planning and inference*, 90(2):227–244, 2000.

[21] J. Snell, K. Swersky, and R. Zemel. Prototypical networks for few-shot learning. In *NeurIPS*, 2017.

[22] M. Sugiyama. Direct approximation of divergences between probability distributions. In *Empirical Inference*, pages 273–283. Springer, 2013.

[23] M. Uehara, I. Sato, M. Suzuki, K. Nakayama, and Y. Matsuo. Generative adversarial nets from a density ratio estimation perspective. *arXiv preprint arXiv:1610.02920*, 2016.

[24] J. Vanschoren. Meta-learning: a survey. *arXiv preprint arXiv:1810.03548*, 2018.