# OpenReview forum: "Meta-Learning for Relative Density-Ratio Estimation"
_NeurIPS.cc/2021/Conference — NeurIPS 2021 Poster_

### Official Review · Reviewer_Q8XR · 2021-07-04

**Rating:** 6
**Confidence:** 4

**Summary:**

The paper shows how one can estimate density ratios when very few data points of the two sets of interested are available. In order to do that, the idea is to leverage other datasets that contain related tasks. The ratio is estimated via a linear model to a latent space that represents each dataset.

**Limitations And Societal Impact:**

The limitations mentioned in "Main Review" were not addressed in the paper; however other limitations were addressed.

**Main Review:**

In general, the paper is well written and the problem is relevant. However, the notation is a bit convoluted and could be improved. For instance, $S_{nu}$ is used to denote target datasets not used for training (Section 4.1), however, the training algorithm itself also uses $S_{nu}$ to denote a subset of one of the training sets.

Major comments:

- The results are encouraging overall, however, I miss comparisons to simpler approaches that would be more appropriate for small datasets than kernel-based and nnets methods. For instance, knn (see Kremer, Jan, et al. "Nearest neighbour density ratio estimation for large-scale applications in astronomy." Astronomy and Computing 12 (2015): 67-72.) and (parametric) probabilistic classification-based ratio methods would provide a more fair baseline for Section "5.2 Relative Density-ratio Estimation".

- As expected, all methods in Table 2 improve their performance as $N_S$ grows. In particular, MMD starts to give better results for $N_S=4$ (even though it does not use the auxiliary datasets). I think it is important to understand what are the limitations of the method. If $N_S$ increases even further, would the other methods that don't use auxiliary datasets be more competitive?

- Regarding the ablation study (especially Table 4): I struggle to understand how your method without latent vectors ("No Latent") is able to learn something meaningful. If the ratio estimator has the shape $w^t h(x)$ (which is what I understand by "no latent"), how can such an estimate adapt to the new datasets? In particular, won't it give the same estimated ratio if you swipe the dataset to be used on the denominator with the one to be used on the numerator? Or are w and h retrained with the support instances of the true target datasets? If the answer is the latter, the text and Algorithm 1 are not clear, and a follow-up question would be whether the AUC results are based on data points not used in this last step.

- I also miss a discussion about the intuition behind the method. For instance, how should the auxiliary datasets relate to the target datasets in order for this procedure to work?

Minor comments:

- Why is NoSadapt-FT worse than NoSadapt? Shouldn't it improve NoSadapt because of the fine-tuning?

- The code has not been submitted (and they say it won't be available because it is proprietary), so it is not possible to check all details about the experiments, nor it is possible to reproduce the examples.

- Why is $N_S$ set to 10 on the toy example but to at most 5 in MNIST and ISOLET?


**Time Spent Reviewing:**

10

---

> ### Author Response · Authors · 2021-08-09
> **Response to Reviewer Q8XR**
>
> We would like to thank the reviewer for the constructive comments and suggestions!
>
> **> the notation is a bit convoluted and could be improved**
>
> We are sorry for some convoluted notations.
> We would like to modify the notations in the revised paper to improve clarity.
>
> **> I miss comparisons to simpler approaches**
>
> Thank you for the great suggestion.
> We evaluated the probabilistic classification-based method with a logistic regression model in your comment (PC), the knn method in your comment (KNN), and RuLSIF with a linear model, $r _{\alpha} (x)=w^{\top} x$ (LRuLSIF).
>
> The averages of the results over different target support instance sizes are as follows:
>
> **Relative DRE**
>
> | data | Ours | PC | KNN | LRuLSIF | PC-FT |
> | ---- | ---- | ---- | ---- | ---- | ---- |
> | Mnist-r | -0.754 | -0.725 | -0.633  | -0.610 | -0.565 |
> | Isolet | -0.895 | -0.873 | -0.776 | -0.613 | -0.568 |
>
> **Dataset comparison**
>
> | data | Ours | PC | KNN | LRuLSIF | PC-FT |
> | ---- | ---- | ---- | ---- | ---- | ---- |
> | Mnist-r | 92.88 | 86.22 | 80.08 | 84.35 | 78.81 |
> | Isolet |98.56 | 97.06| 86.51 | 45.20 | 82.66 |
>
> **Outlier detection**
>
> | data | Ours | PC | KNN | LRuLSIF | PC-FT |
> | ---- | ---- | ---- | ---- | ---- | ---- |
> | IoT | 96.94 | 76.72 | 50.72 | 89.37 | 85.05 |
> | Landmine | 65.31 | 50.47 | 50.19 | 42.17 | 53.19 |
> | School | 63.55 | 54.29 | 52.86 | 53.73 | 54.38 |
>
> Here, PC-FT uses a neural network model, and fine-tunes the last layer of the learned model of PC with source datasets by using target support instances.
> We omitted a fine-tuning method for LRuLSIF since it is equivalent to RuLSIF-FT in the paper. Also, we omitted a fine-tuning method for KNN since its modification is not trivial.
> Our method clearly outperformed these methods. This is because our method is well designed for few-shot relative DRE.
> We would like to add these results in the revised paper.
>
> **>  If $N_S$ increases even further, would the other methods that don't use auxiliary datasets be more competitive?**
>
> Yes. When target support instance size $N_S$ is large, the other methods that do not use auxiliary datasets become more competitive. For example, when $N_S=10$, test AUCs of our method and MMD were 99.7 and 99.8, respectively in the dataset comparison experiment. However, when $N_S$ is small, our method clearly outperformed the other methods as described in Table 2. Since our work focus on relative DRE with a few instances, we believe that our work still has sufficient contributions.
> We will include some discussion for this point in the revised paper.
>
> **>I struggle to understand how your method without latent vectors ("No Latent") is able to learn something meaningful**
>
> As you mentioned, NoLatent uses $r _{\alpha}(x)=w^{\top}h(x)$ as the relative density-ratio model. Although the parameters of neural network $h$ are dataset-invariant, datasets-specific linear weights $w$ can be obtained with support instances by Eq. (5) while fixing $h$. Therefore, NoLatent can perform relative DRE by using the target dataset (support instances) information. The difference between our method and NoLatent is whether latent vectors for datasets are used or not. We will clarify this in the revised paper.
>
> **> a follow-up question would be whether the AUC results are based on data points not used in this last step**
>
> Results in our experiments are calculated on target data, which are not used for our training (Algorithm1.)
> Specifically, in relative DRE experiments, the results were calculated on test target instances, which are different from target support instances, to evaluate the generalization performance of estimated relative density-ratio.
> In dataset comparison experiments, as described in lines 261 to 270,
> the similarity score of two datasets is calculated based on the relative DRE result with target support instances,
> and AUC is calculated on 100 target dataset pairs with these scores.
> In inlier-based outlier detection experiments, as described in lines 298 to 325,
> AUC is calculated on unlabeled target instances, which are also used for estimating relative DRE, by disclosing their labels.
> This transductive setting is the standard setting for inlier-based outlier detection, in which the aim is to find outliers in unlabeled data
> [1,14,18,29,49].
>
> **>I also miss a discussion about the intuition behind the method**
>
> Our method learns how to learn from small data by using multiple datasets. As with other meta-learning methods such as MAML, neural processes, and prototypical networks, shared neural networks such as $h$, $f$, and $g$ are used for accumulating knowledge of how to learn. We described some additional intuition why our framework works well in the first answer to the reviewer r99y. Please see the answer for a better understanding.
> As described in Limitations in the supplemental material, when source and target datasets are significantly different, there is a risk of degrading the performance on the target datasets. We think that it is an important and unsolved research direction to see how similar datasets can prevent this. We would like to pursue this research as well.
>
> **>Why is NoSadapt-FT worse than NoSadapt?**
>
> This is a good question. Dataset-invariant linear weights $w$ of NoSadapt are learned such that NoSadapt can improve the expected test performance by the meta-learning framework.
> However, the learned model of NoSadapt with source datasets is not designed to be fine-tuned. Thus, fine-tuning of $w$ with target datasets might deteriorate performance. This is the reason that NoSadapt often worked better than NoSadapt-FT in our experiments. We will clarify this in the revised paper.
>
> **>Why is  set to 10 on the toy example but to at most 5 in MNIST and ISOLET?**
>
> We describe these results in relative DRE experiments.
> The average results are as follows:
>
> | data | $N_S$ | Ours | RuLSIF | RuLSIF-FT |
> | ---- | ---- | ---- | ---- | ---- |
> | Synthetic | 5 | -0.600 | -0.544 | -0.533 |
> | Mnist-r | 10 | -0.817 | -0.539 | -0.523 |
> | Isolet | 10 | -0.911 | -0.636 | -0.516 |
>
> Our method clearly outperformed the others for each data.

---

> > ### Author Response · Authors · 2021-08-31
> > **Gratitude and Requests**
> >
> > We greatly appreciate your work on our paper.
> >
> > We believe we have addressed all of your concerns and questions in our comments.
> > We would be grateful if you could update your scores if you are satisfied with our comments.
> >
> > If you have any additional questions or concerns, please let us know.
> >
> > Best regards,
> >
> > Authors

---

> > > ### Comment · Reviewer_Q8XR · 2021-09-02
> > > **thanks**
> > >
> > > Thanks for addressing my questions. I'll increase my score to "6: Marginally above the acceptance threshold".

---

### Official Review · Reviewer_Yp9Q · 2021-07-12

**Rating:** 7
**Confidence:** 2

**Summary:**

The authors are proposing to use meta-learning to estimation the density ratio which is a very important problem in machine learning. Density ratio estimation is a very challenging problem as it involves matching an infinite number of statistical moments, which can cause stability issues. The authors have constrained the problem so that they are addressing so that they have limited data from the source domain to estimate the density. The density-ratio is expressed using a permutation-invariant neural network.

The authors derive the loss function for the density ratio that is represented by a neural network. Then show that the parameters of the neural network can be estimated by splitting the data set into support instances and test instances. The authors have demonstrated its capability for relative density ratio estimation with applications to inlier-based outlier detection. Their results have shown to be superior under all matrix.

**Limitations And Societal Impact:**

Yes

**Main Review:**

Overall, the paper is quite well written. The technical part is well explained and well structured. I do think there is some novelty in introducing meta-learning for relative density ratio estimation. My only comment is that there could be stronger experimental results. For example, the authors have compared their results to a number of approach. They have claimed that their approach is a convex optimization, so that there should be no variation in the AUC. However, other approaches may not be convex, it may be good to show some variance in the model after each run, with different initializations or parameters.

**Time Spent Reviewing:**

3

---

> ### Author Response · Authors · 2021-08-09
> **Response to Reviewer Yp9Q**
>
> We would like to greatly thank the reviewer for the positive comments!
>
> **> They have claimed that their approach is a convex optimization, so that there should be no variation in the AUC. However, other approaches may not be convex, it may be good to show some variance in the model after each run, with different initializations or parameters.**
>
> Thank you for the constructive suggestion. Although our optimization problem for the support set adaptation (Eq. 4) is convex w.r.t. $w$, our whole objective function (Eq. 8) is not convex w.r.t. the neural network parameters. Therefore, the results have some variations.
> (Please note that we do not state that there is no variation in the results of our method in the paper.)
>
> In the submitted paper, we have reported average results of 10 runs with different source/validation/target splits for all methods.
> Due to the space limit in the main paper, we did not show variances in all tables.
> Instead, we performed the paired t-test and reported their results, which showed that our method is statistically better than the other methods with $p=0.05$, as described in the captions of Tables 1,2, and 5.
> We would like to include the variances (standard deviations) in the revised paper or the supplemental material.

---

### Official Review · Reviewer_VnfC · 2021-07-13

**Rating:** 4
**Confidence:** 4

**Summary:**

In this paper, a meta-learning method for relative density ratio estimation is proposed. The proposed method uses a permutation-invariant network to learn a network that outputs the relative density ratio of source and target data, and then uses the learned network to output the relative density ratio of a new source-target pair.

**Limitations And Societal Impact:**

Yes.

**Main Review:**

This paper is well motivated and well written so easy to follow.
I acknowledge the importance of relative density ratio estimation, its difficulty in a data-scarce environment, and the idea of meta-learning for addressing the problem. As far as I know, this is the first attempt to solve the relative density ratio estimation problem in the meta-learning framework. It is also highly regarded that the proposed method empirically outperforms conventional methods. However, it is unclear what is mechanism enables to transfer of knowledge.
As briefly mentioned in lines 144 to 146, it is contributed to the sharing of the network structure, but there is no theoretical or experimental evaluation of when/how it will work and when it will not. There are no guarantees in terms of model selection or learning theory.
Overall, the present paper lacks theoretical underpinning and the contribution is not enough for acceptance to this venue.

**Time Spent Reviewing:**

4

---

> ### Author Response · Authors · 2021-08-09
> **Response to Reviewer VnfC**
>
> We would like to thank the reviewer for the positive comments and constructive feedbacks!
>
> **>it is unclear what is mechanism enables to transfer of knowledge ... there is no theoretical or experimental evaluation of when/how it will work and when it will not. Overall, the present paper lacks theoretical underpinning and the contribution is not enough for acceptance to this venue.**
>
> Thank you for the insightful comment. The meta-learning methods including our method learn how to learn from small data using various datasets.
> Specifically, they directly maximize the expected test performance of a model after it has been trained on small data by using various datasets.
> To retain the knowledge of how to learn, representative meta-learning methods such as MAML, neural processes, and prototypical networks, use shared neural networks like our method. The high expressive capabilities of neural networks enable the model to store the knowledge. In fact, many studies have reported that these methods empirically work well [8,10,16,41]. This is an intuitive explanation of the meta-learning methods including our method.
>
> To empirically investigate when/how our method will work and when it will not,
> we performed the ablation study in Tables 3, 4, and 6 in the main paper.
> This study shows that it is important to consider both latent vectors of datasets and linear weights adaptation to support instances simultaneously (i.e., our method) for useful knowledge transfer.
> In addition, we further investigated our method from various perspectives to find out which part of our method is important in Section 4 of the supplemental material.
> For example, in subsection 4.1, we found that relative DRE is more stable and has better performance than DRE in few-shot learning problems, and thus, relative DRE is essential in our framework.
> We think that these results suggest the validity of the design of our method and help to understand our method.
>
> We think that it is important to develop new methods that empirically work well for important problems such as DRE, which is admitted by almost all the reviewers, even if they are not theoretically supported.
> There are many works that have a great impact on subsequent studies such as transformer [a], which has more than 25000 citations, even without theoretical guarantees.
> As admitted by the reviewer r99y, we believe that our work has such potential, and
> will sufficiently contribute to NeurIPS.
>
> [a] Vaswani, Ashish, et al. "Attention is all you need." Advances in neural information processing systems (NeurIPS). 2017.

---

> > ### Author Response · Authors · 2021-08-31
> > **Gratitude and Requests**
> >
> > We greatly appreciate your work on our paper.
> >
> > We have answered your concerns in our comments.
> > We would be grateful if you could update your scores if you are satisfied with our comments.
> >
> > If you have any additional questions or concerns, please let us know.
> >
> > Best regards,
> >
> > Authors

---

### Official Review · Reviewer_r99y · 2021-07-16

**Rating:** 7
**Confidence:** 4

**Summary:**

This paper addresses the estimation of the relative density ratio using only a few instances from two datasets.  The authors propose a meta-learning method in which a neural network is trained to output the embeddings reflect the characteristics of two datasets.  In addition, they proposed to use the closed-solution of a linear model on the embedded space.  The experimental results show that the proposed method outperforms conventional methods on dataset comparison problems and outlier detection when a few instances are available.

**Limitations And Societal Impact:**

The limitations described in the Conclusion and the supplemental material are adequate.


**Main Review:**

Originality:
The few-shot estimation of the relative density ratio is a novel problem setting and valuable for applications.
Although there are attempts to integrate deep neural networks into the estimation of the relative density ratio, the application of meta-learning is also novel for the DNN-based estimation of the relative density ratio.

Quality:
The proposed approach is reasonable, and empirical experiments prove its effectiveness.
The paper also provides an ablation study to confirm the effectiveness of the specific design choices, such as the representation learning of two datasets.

Clarity:
The paper is well-organized.
Although most of the paper is clearly written, integrating the closed-form solution of w into the training of DNN and the estimation of the relative density ratio is not clear enough to implement the proposed method.  I think it will be more clear if the details of NoSadapt and NoSadapt-FT are explained.


Significance:
The few-shot estimation of the relative density ratio is an important problem for real-world applications.   The proposed meta-learning approach can be a baseline for future work.
Since the few-shot setting in the experiments is artificial, I would like to see a real-world application.

**Time Spent Reviewing:**

4

---

> ### Author Response · Authors · 2021-08-09
> **Response to Reviewer r99y**
>
> We would like to greatly thank the reviewer for the positive comments!
>
> **> integrating the closed-form solution of w into the training of DNN and the estimation of the relative density ratio is not clear enough. I think it will be more clear if the details of NoSadapt and NoSadapt-FT are explained**
>
> NoSadapt uses $r_{\alpha}(x)= w^{\top} h([x,z_{{\rm nu}},z_{{\rm de}}])$ as the relative density-ratio model. Here, both linear weights $w$ and neural network parameters of $h$ are dataset-invariant, i.e., they are usual trainable parameters and are trained with source datasets.
> Latent vectors $z_{{\rm nu}}$ and $z_{{\rm de}}$ are used to treat dataset-specific information.
> NoSadapt performs relative DRE on target datasets only by inferring $z_{{\rm nu}}$ and $z_{{\rm de}}$ from the target support instances.
> However, it might be difficult to perform accurate relative DRE since
> it does not explicitly minimize the squared error of relative DRE on the target support instances Eq. (3).
>
> NoSadapt-FT further fine-tunes linear weights $w$ of the learned model of NoSadapt by minimizing the squared error of relative DRE on the target support instances to improve the estimation performance. As a result, we obtain Eq. 5. However, it might be also difficult to improve the performance since the learned model with source datasets is not designed to be fine-tuned.
>
> In contrast, our method directly improves the expected test performance of the fine-tuning with support instances using the meta-learning framework.
> Specifically, the fine-tuned model with support instances (Eq. 6) is evaluated on query instances (Eq. 8) in source datasets, and its loss can be backpropagated with standard gradient descent methods to improve the expected test performance. This can be performed since closed-form solution ${\hat w}$ is differentiable w.r.t the neural network parameters.
> We note that inverse matrix operations, which are used in Eq. 5, are implemented in standard neural network libraries such as PyTroch.
> Our method performs relative DRE on target datasets by both estimating latent vectors of datasets and adapting linear weights to the target datasets.
> For another perspective, we can regard ${\hat w}$ as a neural network layer, and all the neural networks including this special differentiable layer are trained by minimizing the loss on query instances (Eq. 8) as standard neural network training.
> We would like to add these explanations in the proposed method section and the experiment section in the revised paper to improve clarity.
>
> **> Since the few-shot setting in the experiments is artificial, I would like to see a real-world application**
>
> Thank you for the good comment. We admit that the relative DRE experiments are a bit artificial. However, we included the experiments to demonstrate that our method can improve relative DRE performance with a few instances. The outlier detection experiments with real-world benchmark data described in subsection 5.4, which are also used in a recent NeurIPS paper [23], demonstrate that our method is quite useful for outlier detection, which is an important problem in practice.
> Since DRE can be used for various real-world applications such as domain adaptation and change point detection, we would like to evaluate our method in such real-world applications.

---

### Decision · Program_Chairs · 2021-09-27

**Decision:**

Accept (Poster)

**Comment:**

The reviews are mostly positive and the general sentiment is that the meta-learning approach (for estimating density ratios when there are few instances) is well-motivated and the results encouraging. There were some concerns of lack of theoretical underpinnings (Vnfc), the method limited in terms of applications (Q8XR), and experiments being artificial (r99y). The authors need to address the last two concerns in a revision, and include the additional experimental support in e.g. Supplementary Materials.